# BETTER STEP, A FORMAT AND DATASET FOR BOUNDARY REPRESENTATION

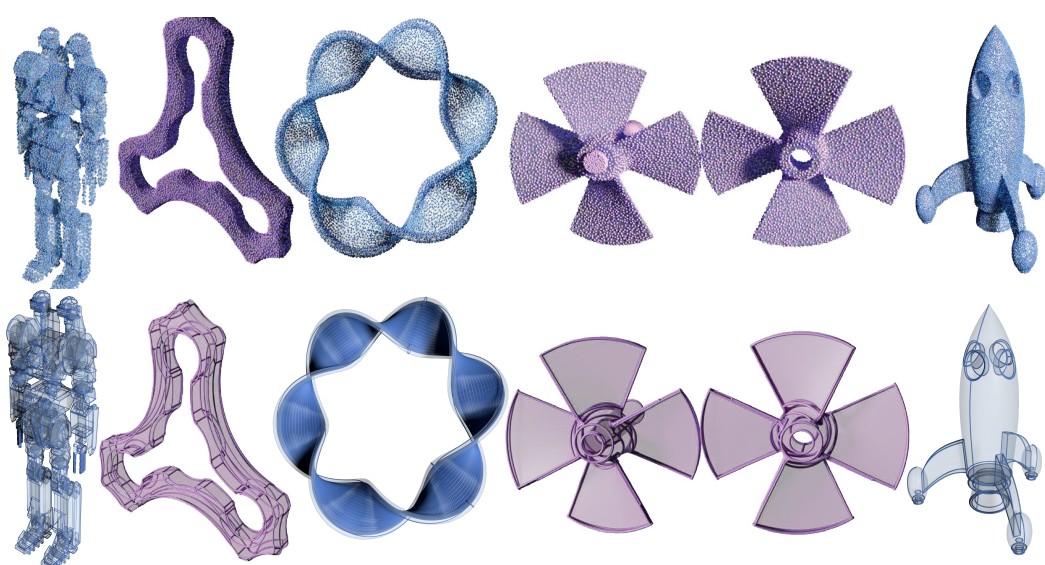

Figure 1: Examples of a few B-reps from our different datasets. We randomly sample (top) the B-reps (bottom) with our library.

## ABSTRACT

Boundary representation (B-rep) serves as the primary format for 3D geometry in computer-aided design (CAD), integrating parametric geometry with explicit topology to model complex components and assemblies. Despite its omnipresence, research in machine learning rarely leverages B-reps directly; instead, STEP files are typically parsed with (proprietary) kernels and reduced to meshes, point clouds, or basic face-edge-vertex graphs. These simplifications discard important details (e.g., parametric patches or topology) present in B-reps and introduce additional challenges related to licensing, compatibility, and scalability.

We introduce Better STEP, an open source format that preserves the full fidelity of B-reps while enabling direct, efficient access in standard ML frameworks, removing dependence on proprietary software. In addition, we introduce an open-source Python library for querying and processing B-rep data. The Python package provides standard functionalities for querying geometry (e.g., surface sampling, normal estimation, curvature computation), as well as topological structure, thereby facilitating integration into existing pipelines.

To demonstrate the effectiveness of our format, we converted the Fusion 360 and ABC datasets, comprising over one million CAD models Koch et al. (2019); Lambourne et al. (2021); Willis et al. (2021a;b). We further showcase the universality of our Python package by generating test data for four representative downstream tasks; these experiments did not require fine-tuning or modifying the original models, underscoring the ease with which our data can be integrated into existing machine learning workflows.

# 1 Introduction

Boundary representation (B-rep) is one of the most common formats for representing 3D shapes in solid modeling and computer-aided design, and it is widely used in industry due to its ability to describe precise and complex geometries. B-rep represents shapes as a collection of intersecting parametric surfaces, allowing for the definition of complex smooth surfaces. In recent years, several large datasets have been created containing thousands of B-reps in STEP format Koch et al. (2019); Lambourne et al. (2021); Willis et al. (2021a;b).

Despite the availability of large datasets, B-reps are typically stored in STEP format, which requires proprietary CAD kernels for access and processing. Furthermore, different kernels and different kernel versions are incompatible. This challenge has led to the flourishing development of solutions such as CADFix and CADDoctor, whose primary function is to repair and convert STEP files across different kernels and versions. These barriers affect B-rep usage in large learning pipelines, as they limit the possibility of deploying them on computing clusters due to the formats' highly unstructured and undocumented nature.

Therefore, while the primary contribution of this work is the introduction of a dataset and query library for B-reps, the applicability of this resource extends well beyond the scope of the present paper. Recent generative approaches illustrate the potential of integrating large language models (LLMs) with textual or visual inputs for CAD generation. For instance, Xu et al. (2024); Jayaraman et al. (2022) introduces the Furniture B-rep dataset, which is constructed by hand-picking furniture CAD models. Alternatively, Li et al. (2025); Rukhovich et al. (2024) utilize a code representation to generate CAD models. However, their applicability is limited by the small size of the dataset (for the furniture) or by the expressiveness of the code. Our dataset and format will bridge the gap, enabling LLM to generate complex B-rep models fully.

This paper introduces an alternative equivalent format, an open-source library to process it, and a corresponding dataset (Figure 1) for STEP files. Our format is fully specified (Appendix A) and it is based on the standard half-edge format. To foster cross-language and cross-platform compatibility, we encode it as a dictionary using the HDF5 format; with our format, any application can read and process the data. To ease integration in existing pipelines, we provide a Python package with standard functionalities such as sampling, normals, or curvature. We convert the Fusion 360 and ABC datasets.

To show the effectiveness of our format and library, we use our library on a series of common learning tasks (i.e, normal estimation, denoising, surface reconstruction, and segmentation), showing how easy it is to use; we confirm that the accuracy obtained by every method is inline with the results reported by the authors. We note that for the classification task in Fu et al. (2023), the authors used the triangle meshes in the ABC dataset and used a heuristic to retrieve the parametric information (as it is lost in the meshes). With our library and format, this information is naturally and easily obtainable.

We hope that our dataset and format will become the new standard benchmark for learning tasks on 3D shapes and that the ability to retrieve parametric information will lead to new, exciting discoveries and progress.

# 2 Related Work

Applying machine learning to 3D geometry has created a growing demand for large, richly annotated datasets of 3D shapes in formats that preserve geometric fidelity and support editability. Early shape datasets (Chang et al. (2015); Zhou & Jacobson (2016); Qi et al. (2016)) primarily contained annotated meshes or point clouds and were the main drivers of data-driven research on 3D shape understanding and processing. As research in geometric deep learning and computer-aided design (CAD) has progressed, there has been a growing need for representations that go beyond discrete approximations. Recent advances in these fields emphasize the importance of using continuous and smooth B-reps and parametric surfaces Fu et al. (2023); Dupont et al. (2022); Cheng et al. (2024); Jayaraman et al. (2021). B-reps are composed of trimmed parametric surfaces and explicitly define the adjacency relationships that connect them into a coherent solid Lambourne et al. (2021). They offer the advantage of richer semantic annotations (e.g., the type and shape of components, and

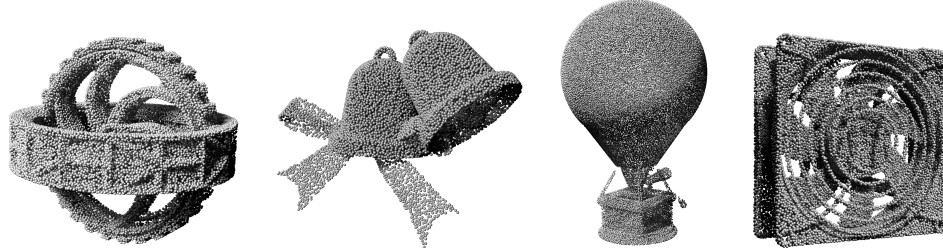

Figure 2: Point-cloud for a model where OpenCascade fails to generate a mesh.

how they are assembled), enabling learning tasks that incorporate both geometry and the generative design process.

Unlike mesh-based representations, which discretize geometry and may lose important structural information, B-reps retain the exact geometry and topology defined during CAD modeling Willis et al. (2021b). This feature enables precise querying and captures high-level design semantics, making B-reps well suited for applications such as reconstruction Zhang et al. (2024) and constraint inference Cheng et al. (2024). Furthermore, since multiple CAD models can result in identical sampled meshes, mesh-based data tends to be more ambiguous. In contrast, B-reps provide a more reliable foundation for tasks requiring interpretability and reversibility Dupont et al. (2022); Wu et al. (2021).

Recognizing these advantages, recent efforts have focused on building datasets natively supporting B-reps or CAD formats such as STEP. ABC Koch et al. (2019) was among the first to collect one million 3D STEP files. This dataset sparked growing interest in developing more datasets that support native B-reps Lambourne et al. (2021); Jayaraman et al. (2021), enabling the design of neural architectures that operate directly on these structures rather than on their triangulated approximations. Subsequently, the Fusion 360 Gallery Willis et al. (2021a;b) dataset introduced thousands of STEP file sequences, along with the sequence operations used to construct the final model.

As a result, this new perspective has inspired the creation of additional datasets and benchmarks, such as DeepCAD Wu et al. (2021), which provides over 170,000 models with construction sequences. Brep2Seq Zhang et al. (2024) introduces a large-scale collection of auto-synthesized, feature-based CAD models. More recently, datasets such as Param20K Cheng et al. (2024) have enriched parametric data with explicit annotations.

## 3 LIBRARY

We developed two Python libraries to support our data workflows: STEPToHDF5, for dataset conversion, and ABS, for data processing. Both libraries use the HDF5 format in combination with NumPy for efficient binary encoding. We chose HDF5 for its stability, cross-platform support, and compatibility with multiple languages (e.g., Python, C++, Julia, R, MATLAB). While formats like Pickle may yield smaller files, their reliance on specific languages and Python versions limits long-term usability.

### 3.1 STEPTOHDF5

STEPTOHDF5 uses OpenCascade CASCADE to parse and extract the geometric and topological information from the STEP file and convert it into our dictionary-based format (Appendix A). In our format, each file is decomposed into a sequence of parts, where each part contains its own geometry, topology, and mesh. There are no topological connections across parts, which reflects the characteristic of the STEP format design, where assemblies do not encode topological relationships between components.

The geometry includes all parametric representations, such as curves and surface patches, while the topology defines the connectivity between these geometric entities, specifying how edges, faces, and

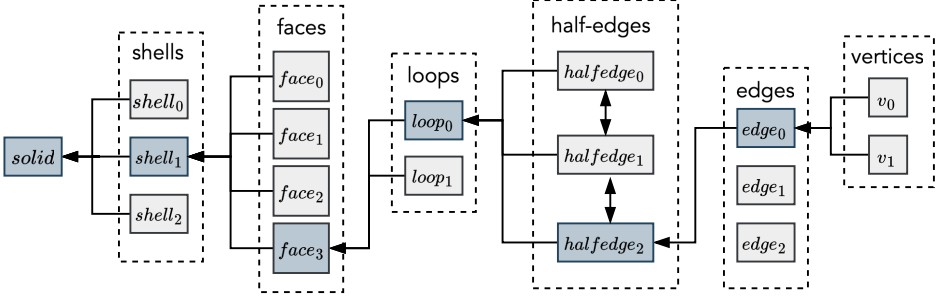

Figure 3: Hierarchical structure of our format. From the root structure (solid) to the leaf (vertices).

shells are assembled into a coherent structure. We note that the meshing algorithm in OpenCascade is not fully robust, and approximately 5% of the models fail to produce a mesh (Figure 2).

**Geometry.** The geometry encodes geometric information in the form of a list of two- and three-dimensional curves and surfaces and a matrix of vertices. Each geometry entity has its own parametric domain (a line for curves and a rectangle for surfaces) and the parameters that define it. For instance, a B-spline curve has a set of control points and knots, whereas a plane has two axes and a location. A detailed specification of the geometry representation is provided in (Appendix A).

**Topology.** The topology contains a hierarchical structure of the STEP file (Figure 3). To save space and maintain consistency, we store only the top-down relations (e.g., a face contains loops, but a loop does not store the face it belongs to). When needed, the library provides an API to recover reverse links. In topology, each part contains one or more independent solids. The solid is the root node that contains one unique field to store the list of shells. Every shell lists its faces and an orientation flag. For manifold solids the flag is always true. In the case of a non-manifold cell complex, where multiple shells may share one face, the flag represents whether the face normal needs to be flipped to point outwards from the solid volume the shell bounds. Both solid and shell are purely topological entities and do not have a geometric counterpart.

A face represents a surface patch and contains a surface index that points to the corresponding surface in the geometry. In addition, it includes an orientation flag, one outer loop, and zero or more inner loops. Each loop is a closed poly curve that represents the trimming of the face. It consists of a list of half-hedges that can be shared by two edges. A half-edge includes the index of a two-dimensional curve in the geometry, its mates (the opposite half-edge), an orientation flag, and the associated edge index. As the edge contains the pointer to the three-dimensional curve in the geometry, the orientation flag is used to properly orient the half-edge in the loop.

**Mesh.** The hierarchical structure of the mesh is organized by face groups. Each subsection of the mesh has a unique numeric identifier (e.g., `000` or `001`) that corresponds to the face indices in the topology. This structure inherently preserves the connection to the topological data, enabling access to all semantic labels.

## 3.2 ABS

Our format only contains equivalent information to the B-rep data, and using it directly in an application might be challenging. We developed a library that allows processing, navigating, sampling, and extracting features from the dataset to facilitate its usage. ABS allows reading and navigating the HDF5 files as a standard half-edge data structure. We provide a simple `read_parts` to read all parts present in the file.

**Topology Navigation.** ABS supports extraction and navigation of information regrading topological attributes of the B-rep, such as edge degrees, or face adjacencies. Each topological entity (faces, edges, loops, etc.) is represented by its own class with dedicated query methods. For instance, Listing 1 shows how to retrieve adjacent faces or access the first loop of a face.

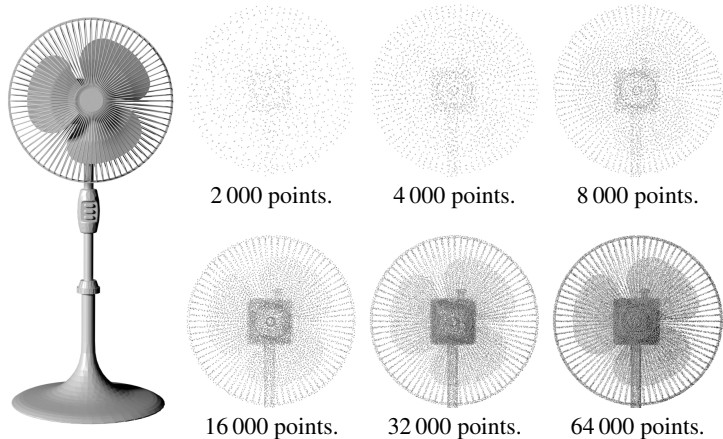

Figure 4: Example of a complex fan model (left) sampled with an increasing number of points. As the resolution increases, the small details become visible.

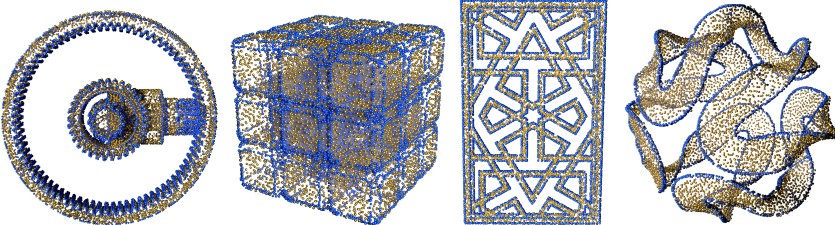

Figure 5: Example of point clouds sampled using Listing 2; we highlight the feature edges in blue.

```python
from abs import read_parts

parts = read_parts(file_path)
current_face = parts[0].Solid.faces[1]
adj_face_list = current_face.find_adjacent_faces()
current_loop = current_face.loop[0]
```

Listing 1: Example of finding adjacent faces for the second face in the first part.

**Geometry Sampling.** ABS can generate random points sampled directly from the *continuous parametric* shapes with an arbitrary number of sample points, and evaluate parametric derivatives (Figure 4). We provide a simple function sample_parts that uses a lambda function to decide which information to extract (Section 5 shows more concrete examples). The lambda function has access to the current part, the current topological entity (either a face or an edge), and the random points in the parametric domain. Its responsibility is to return data associated with the points (e.g., a normal or label), or None if the entity must be skipped. Listing 2 shows a typical example of how to use ABS, we use read_parts to read the file and compute_labels and sample_parts to sample the shape and obtain a binary label to mark feature edges or patches (Figure 5).

```python
from abs import read_parts, sample_parts

def compute_labels(part, topo, points):
    if topo.is_face(): return 1
    else: return 0

parts = read_parts(file_path)
P, S = sample_parts(parts, num_samples, compute_labels)
```

Listing 2: Example of computing normal at every point.

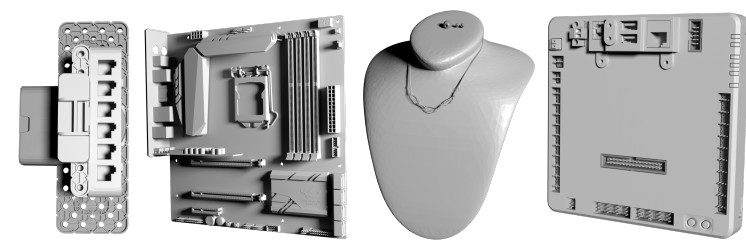

Figure 6: Example of meshes of models with thousands of patches.

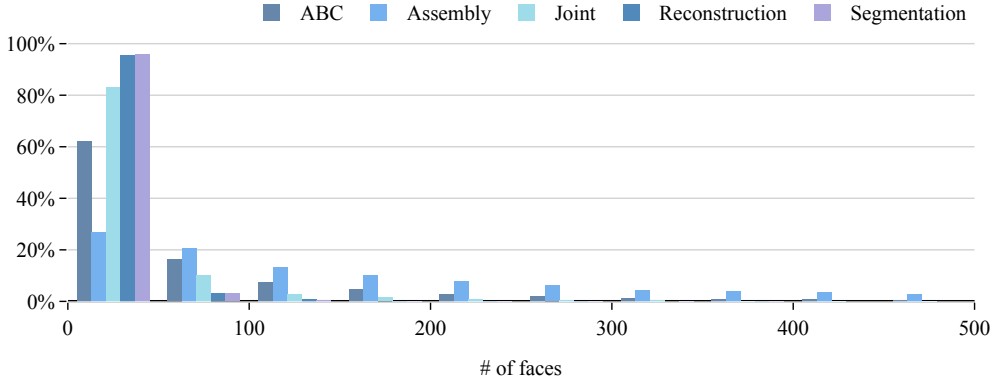

Figure 7: Distribution of of faces per model for the different datasets.

**Mesh Extraction.** Since the mesh is not connected with the topology and the geometry of the B-rep, we provide a utility function `read_meshes` to retrieve the meshes as point-triangle dictionary, one per part per face (Listing 3). Note that, as not every face can be meshed, the pair can be `None`. For instance, `meshes[0][9]` contains the mesh of the tenth face of the first part. Additionally, we have a simple function that concatenates every meshed patch into a unique, consistent mesh.

```
from abs.utils import read_meshes, get_mesh

meshes = read_meshes(file_path)
V, F = get_mesh(meshes)
```

Listing 3: Example of extracting the mesh from a file.

## 4 DATASET

Our dataset includes one million models from ABC Koch et al. (2019), as well as the Assembly (8 251 models, 16 2707 parts), Joint (23 029 parts), Reconstruction (27 958 parts), and Segmentation (35 680 parts) from Fusion 360 dataset Willis et al. (2021a;b). We converted the data on cluster nodes equipped with Intel E5 v4 Broadwell @ 2.2GHz CPUs. On average, converting a single model takes a few seconds, and processing the entire dataset requires approximately one CPU year. We computed statistics on a random selection of 4 000 models for ABC, on the assembled models for Assembly, and on the entire dataset for Joints, Reconstruction, and Segmentation.

On average, the models contain 137 patches (ABC: 236, Assembly: 590, Joint: 37, Reconstruction: 15, and Segmentation: 15) with models with more than 30 000 patches (Figure 6). The different datasets contain models of varying sizes (Figure 7); Assembly is the largest overall (even though ABC includes the model with the most faces), while Segmentation is the smallest. All models in the dataset contain about 50% planes; with the Reconstruction dataset having the highest proportion at 72% (Figure 8, top). Only the ABC dataset includes a small number of offset, revolution, and extrusion surfaces. Similarly, most models consist primarily of lines and circles, while the Segmentation dataset includes the highest number of unrecognized curves marked as "other" (Figure 8, bottom).

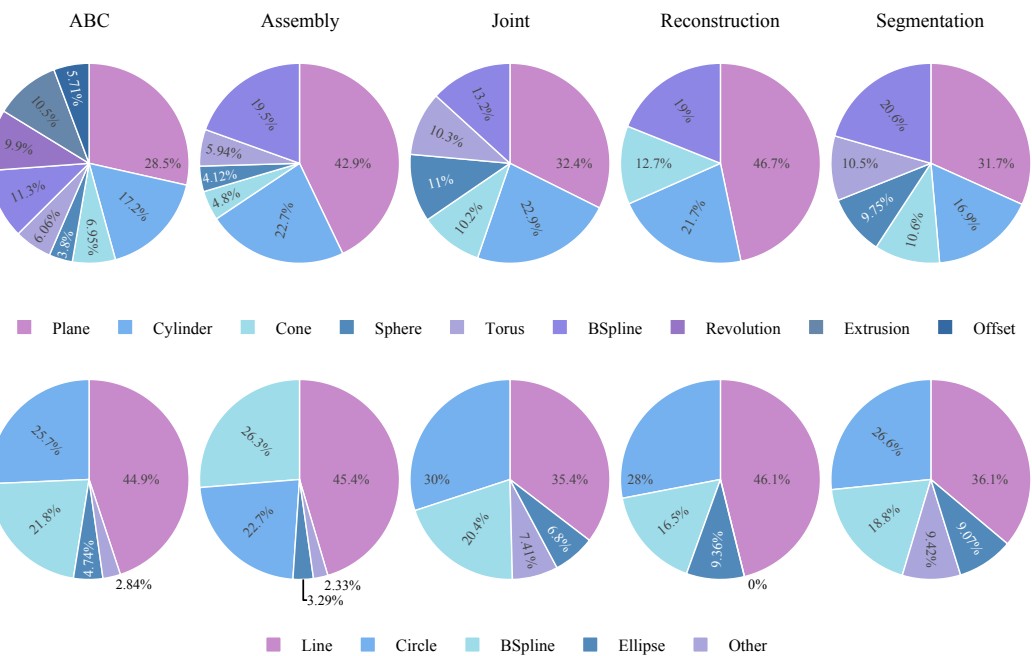

Figure 8: Ratio of faces (top) and curves (bottom) types across the different datasets.

For the Fusion Joint dataset, on average processing a file takes only a few milliseconds (approximately 0.38 seconds for a batch of 1,000 samples). For Fusion Assembly, which contains larger files, the conversion increases to around 0.75 seconds, and for the ABC dataset, which includes the largest files, the average time is approximately 3.57 seconds per file.

The meshing algorithm in OpenCascade is not robust and occasionally fails (i.e., some of the patches have no mesh), with a failure rate of 1.56% for the ABC dataset, 8.82% for the Assembly dataset, 1.04% for the Joint dataset, 0.02% for the Reconstruction dataset, and 0.07% for the Segmentation dataset.

## 5 USE CASES

We showcase the simplicity and versatility of our library by generating data for four point-cloud-based machine learning tasks: normal estimation, denoising, reconstruction, and segmentation. For all use cases, we use the same code as in Listings 2 except that we write a task-specific lambda function. Note that we do not fine-tune or retrain the models; we evaluate them directly using our dataset.

**Normal estimation.** A classical learning problem involves estimating normals from a point cloud, which requires a dataset of point clouds paired with ground truth normals. This can be easily computed from our dataset using the function in Listings 4. We evaluate the DeepFit model Ben-Shabat & Gould (2020) on 8,000 points generated with ABS, sampled from 200 randomly selected models in the ABC dataset. Although the model was trained on piecewise linear geometries (i.e., meshes), it performs well in estimating smooth normals. The percentage of good points (PGP), ignoring normal orientation, is 65.73%, 79.45%, and 90.28% for angular thresholds of $5°$, $10°$, and $30°$, respectively.

```
1 def compute_normals(part, topo, points):
2     if topo.is_face(): return topo.normal(points)
3     else: return None # No normals for edges
```

Listing 4: Extracting the normals.

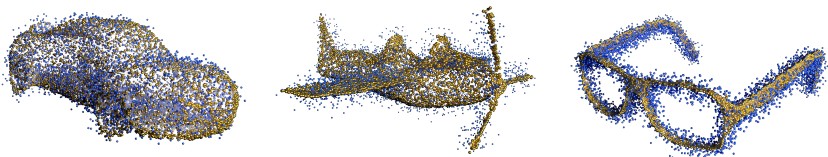

Figure 9: Example of denoising point cloud using PathNet Wei et al. (2024).

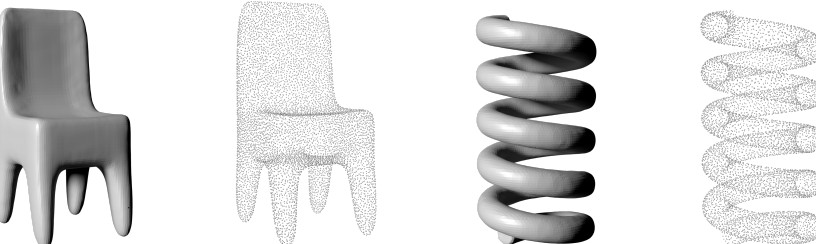

Figure 10: Example of reconstructed surfaces from randomly sampling our dataset using NKSR.

**Denoising.** We use the recently published PathNet Wei et al. (2024) to denoise point clouds. The model consists of a two-stage deep and reinforcement learning pipeline. It dynamically selects the optimal denoising path for each point using a reinforcement learning-based routing agent that adapts to local noise levels and geometric complexity. The model requires only the input noisy point cloud and outputs the denoised result for both training and evaluation (Figure 9). Ground truth can be generated using Listings 5, with noise added afterward. While Wei et al. (2024) train their model using mesh-sampled data, our dataset and library allow direct sampling from smooth parametric surfaces. Despite this slight difference, the method performs comparably when applied to our dataset. We selected 1 000 models from the Assembly dataset, sampled each with 6 000 random points, and added varying levels of Gaussian noise. To evaluate performance, we sampled each model with 10 000 points and computed the MSE (in units of $\times 10^{-3}$) at different noise levels: 33.9, 34.07, and 35.79 for noise levels of 0.5%, 1%, and 1.5%, respectively.

```python
def get_points(part, topo, points):
    if topo.is_face(): return 1 # Return dummy value
    else: return None
```

Listing 5: Extracting just the points.

**Surface Reconstruction.** We selected Neural Kernel Surface Reconstruction (NKSR) Huang et al. (2023) as an example method for reconstructing meshes from a potentially noisy point clouds (Figure 10). This approach represents surfaces as a zero-level set of a neural kernel field fitted to oriented point clouds via a gradient-based energy formulation, using only points and corresponding normals for supervision for training. Both training and evaluation datasets can be generated with the same code as in Listings 4, using denser sampling for training. For our experiments, we selected 1 000 models from the Assembly dataset, sampled these models with varying numbers of points, and added random Gaussian noise. To evaluate reconstruction quality, we computed Chamfer distance and F-Score metrics between the reconstructed surfaces and a dense sampling (15 000 points) of the parametric surfaces. Our findings, summarized in Table 1, indicate slightly lower reconstruction quality compared to results reported by Huang et al. (2023). This reduction in quality is likely due to our direct sampling of parametric surfaces instead of using pre-existing meshes.

**Segmentation.** A complex problem consists of correctly labelling points in a point cloud based on the geometric primitive. For instance, it automatically detects which points belong to a plane or a cylinder. We can use our library to compute the labels as we sample the surface, using a different callback that converts the surface type into the label (Listings 6). We use the BPNet Fu et al. (2023) model that uses labelled points as input. We note that Fu et al. (2023) originally used the meshes in the ABC dataset Koch et al. (2019) and had to recover the patch information and degrees with a heuristic (Fu et al., 2023, Section 4.1); by using our library, this information is readily available

Table 1: Evaluation metrics for surface reconstruction with varying sample sizes and noise levels ($\sigma$). Chamfer distance ($d_C$) values are scaled by $10^3$.

| | 4000 samples | | | 6000 samples | | | 8000 samples | | |
|---|---|---|---|---|---|---|---|---|---|
| | $\sigma = 0$ | $\sigma = 0.005$ | $\sigma = 0.025$ | $\sigma = 0$ | $\sigma = 0.005$ | $\sigma = 0.025$ | $\sigma = 0$ | $\sigma = 0.005$ | $\sigma = 0.025$ |
| $d_c$ | 3.39 | 1.91 | 2.02 | 3.42 | 1.48 | 3.75 | 1.63 | 2.20 | 2.56 |
| F-Score | 84.62 | 85.61 | 60.0 | 87.02 | 84.56 | 61.82 | 85.81 | 85.59 | 63.0 |
| Precision | 82.09 | 82.10 | 52.65 | 83.78 | 79.56 | 54.20 | 81.11 | 80.79 | 55.01 |
| Recall | 91.13 | 92.23 | 72.85 | 93.34 | 94.31 | 75.17 | 95.16 | 95.16 | 77.25 |

Table 2: Accuracy, primitives and times across different noise levels ($\sigma$) for recovering patch degrees using BPNet.

| Noise Level ($\sigma$) | Accuracy | Number of Primitives | Inference Time |
|---|---|---|---|
| $\sigma = 0$ | 85.78 % | 23 | 1.63 |
| $\sigma = 0.05$ | 85.59 % | 24 | 1.53 |
| $\sigma = 0.1$ | 83.89 % | 27 | 1.65 |

as it maintains the B-reps and directly samples the parametric surfaces. We selected 1 000 random assembly parts from the Assembly dataset and sampled and labeled them with 6,000 points. Table 2 shows that the results of the model using ABS on a different dataset are consistent with the data reported by Fu et al. (2023).

```python
def find_primitive_degrees(part, topo, points):
    if not topo.is_face(): return None

    normal = topo.normal(points)
    shape_name = topo.surface.shape_name

    if shape_name == 'BSpline':
        if topo.surface.u_rational or topo.surface.v_rational:
            return None # BPNet only labels Bezier patches
        degree = (topo.surface.u_degree, topo.surface.v_degree)
    elif: shape_name == 'Plane': degree = (1, 1)
    elif: shape_name == 'Sphere': degree = [(2, 2), (3, 3)]
    else: degree [(2, 3), (3, 2)]

    return [normal, degree]
```

Listing 6: Getting normals and primitive degrees.

## 6 CONCLUSION

We introduced a new open, cross-platform, and cross-language format equivalent to a B-rep, along with a Python library based on OpenCascade to convert STEP files, and a library to process the resulting format. We hope our format and library will become the new standard representation for CAD and B-rep processing and machine learning on parametric surfaces. We envision pipelines where our format serves as a bridge between CAD software and state-of-the-art research or where LLM are capable to generate B-rep directly.

While we have already converted several million models, more datasets remain, and we hope that the community will join the effort. Additionally, our conversion algorithm is based on OpenCascade, the only open-source CAD kernel; however, the format itself does not depend on it. Since different STEP files require different kernels, we believe our set of tools can be extended to support other (including commercial) CAD kernels.

Finally, all the models in our dataset are fully rigid; it would be interesting to extend our dataset and format to deformable B-reps.

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

## A   FILE FORMAT

The root of the HDF5 file includes one unique group called parts and has one string attribute `version` (currently version 2.0). The part group contains as many sub-groups as the model has parts, called `part_<n>`. Each `part_<n>` group contains *three* groups: `geometry`, `topology`, and `mesh`.

### A.1   GEOMETRY.

Geometry contains the list of 2D/3D curves, the surfaces, the dataset of vertices and the bounding box of the model.

**Curves.** Each curve can be either a circle $C$, an ellipse $E$, a line $L$, a b-spline, or an Other. All curves contain the `type` (a string encoding the name), an `interval` (the parametric space), and a `transform` (encoded as a $3 \times 4$ matrix in homogenous coordinates) for 3d curves. The parameterization for a curve in $\mathbb{R}^N$ are

$$L(t) = l + t\,d$$
$$C(t) = l + r(\cos(t)a_x + \sin(t)a_y)$$
$$E(t) = (f_1 + f_2)/2 + r_M \cos(t)a_x + r_m \sin(t)a_y,$$

where $l \in \mathbb{R}^N$ is the `location`, $d \in \mathbb{R}^N$ the `direction`, $r \in \mathbb{R}$ the `radius`, $a_x \in \mathbb{R}^N$ the `x_axis`, $a_y \in \mathbb{R}^N$ the `y_axis`, $f_1 \in \mathbb{R}^N$ the `focus1`, $f_2 \in \mathbb{R}^N$ the `focus2`, $r_M \in \mathbb{R}^N$ the `maj_radius`, and $r_m \in \mathbb{R}^N$ the `min_radius`. For a b-spline, we store the `poles` (control points) and `knots`; if it is `rational`, we have the `weights`. We also track if the curve is `periodic` or if it `closed`. Finally, we keep track of the `degree` and the `continuity` of the curve.

**Surfaces** Surfaces can be either a Plane $P$, Cylinder $C_y$, Cone $C_n$, Sphere $S$, Torus $T$, BSpline, Extrusion, Revolution, or Offset. All surfaces contain `trim_domain` (the two-dimensional para-

metric domain), a `transform`, and a `type`. The parameterizations are

$$P(u,v) = l + ua_x + va_y$$
$$C_y(u,v) = l + r\cos(u)a_x + r\sin(u)a_y + va_z$$
$$C_n(u,v) = l + (r + v\sin(\alpha))(\cos(u)a_x + \sin(u)a_y) + v\cos(\alpha)a_z$$
$$S(u,v) = l + r\cos(v)(\cos(u)a_x + \sin(u)a_y)r\sin(v)a_z$$
$$T(u,v) = l + (r_M + r_m\cos(v))(\cos(u)a_x + \sin(u)a_y) + r_m\sin(v)a_z,$$

where $l \in \mathbb{R}^3$ is the `location`, $a_x \in \mathbb{R}^3$ the `x_axis`, $a_y \in \mathbb{R}^3$ the `y_axis`, $a_z \in \mathbb{R}^3$ the `z_axis`, $r \in \mathbb{R}$ the `radius`, $\alpha \in \mathbb{R}$ the `angle`, $r_M \in \mathbb{R}^N$ the `max_radius`, and $r_m \in \mathbb{R}^N$ the `min_radius`. For a b-spline, we store the `poles` (control points), `u_knots` and `v_knots`; if it is `u_rational` or `v_rational`, we have the `weights`. We also track if the curve is `u_periodic`/`v_periodic` or if it is `u_closed`/`v_closed`. Finally, we keep track of the `u_degree`, `v_degree`, and the `continuity` of the curve.

Extrusion $E$ and Revolution $R$ contain a parametric `curve` $\gamma$ following the same standard curve definition.

$$E(u,v) = \gamma(u) + vd$$
$$R(u,v) = \mathcal{R}_a(u)(\gamma(v) - l) + l$$

where $d \in \mathbb{R}^3$ is the `direction`, $l \in \mathbb{R}^3$ is the `location`, and $\mathcal{R}_a \in \mathbb{R}^{3\times3}$ is a rotation matrix round the axis `z_axis`.

Finally, the Extrusion contains another `surface` which can be any of the surfaces and `value`. The surface is defined by extruding the point by `value` along the `surface` normal.

## A.2 TOPOLOGY.

Topology contains 6 groups: `edges`, `faces`, `halfedges`, `loops`, `shells`, and `solids`. All groups contain numerical subgroups, one for every entity. For instance, `/parts/part_001/topology/solids/001` represents the second solid for the first part and `/parts/part_001/topology/halfedges/003` the fourth half-edge.

**Solids.** Each numerical subgroup represents one per solid in the model, each storing one dataset `shells` containing the shell indices. Note that some models have no solid as they are made of only shells; in that case, the solid group has no sub-groups.

**Shells.** Each shell has two datasets: `faces` and `orientation_wrt_solid`; the faces contain face indices, and the orientation boolean flag is used to determine the orientation of the shell. If the flag is false, the orientation of the shell must be flipped.

**Faces.** Each face has `exact_domain`, `has_singularities`, `loops`, `nr_singularities`, `outer_loop`, `singularities`, `surface`, and `surface_orientation`. `Exact_domain` has the exact UV bounds of all loops on the face. The loops contain the indices of the loops in the face, and the outer loop is the index of the loop that contains all other loops. We also record the number of singularities (if any) and their location in the singularities group. The surface containing the index of the geometric parametric surface attached to this face. Similarly to the shells, the orientation of the face is decided by the orientation flag.

**Loops.** Each loop contain one unique dataset `halfedges` containing indices to the half-edges.

**Half-edge.** Every half-hedge has `2dcurve`, `edge`, `mates`, and `orientation_wrt_edge`. The 2d curve is an index for the *geometric* 2d curve, while the edge and mates points to the *topological* edges. Since multiple loops might share edges, the orientation flag indicates if the curve requires flipping.

**Edge.** The edge is the leaf of the tree that contains only pointers to the geometry: `3dcurve` to a 3d curve, and `start_vertex` and `end_vertex` to vertices.

### A.3 MESH.

The mesh group is divided into numerical subgroups, one for each face, with each subgroup storing two datasets: `points` and `triangles`, which define the mesh. For instance, `/parts/part_001/mesh/003/points` and `/parts/part_001/mesh/003/triangle` contain the mesh for the fourth patch of the first part. If a face has no mesh, the corresponding `points` and `triangles` datasets will be empty.

## B    REPRODUCIBILITY STATEMENT

The dataset and library presented in this work will be made public upon acceptance. We can not release or anonymously share the dataset since it is several terabytes.

## C    USE OF LARGE LANGUAGE MODELS

We used LLMs for text editing and polishing and to help retrieve and discover parts of the related work.

