# OpenReview forum: "Better STEP, a format and dataset for boundary representation"
_ICLR.cc/2026/Conference — Submitted to ICLR 2026_

### Official Review · Reviewer_oVFQ · 2025-10-27

**Soundness:** 2
**Presentation:** 2
**Contribution:** 2
**Rating:** 4
**Confidence:** 2

**Summary:**

The paper introduces Better STEP, an ML-friendly HDF5 schema that preserves exact B-rep geometry/topology with optional per-face meshes. It releases STEPTOHDF5 (conversion) and ABS (sampling/labeling), converts >1M CAD models, and demonstrates plug-and-play data generation for normals, denoising, reconstruction, and primitive/degree segmentation using existing models. The core contribution is practical infrastructure that standardizes CAD access and reduces reliance on proprietary kernels, enabling scalable, reproducible ML on B-reps.

**Strengths:**

1. This work provides highly practical infrastructure by offering an ML-ready representation that avoids lossy mesh/point conversions and preserves parametric detail and topology.
2. It integrates easily into pipelines through simple Python APIs for sampling, labeling, and per-face meshes, which reduces data-preparation friction.

**Weaknesses:**

1. Although the paper presents a new format and dataset to facilitate machine learning in the B-rep domain, it includes no experiments on “learning this new representation.” It is unclear to me whether this paper falls within ICLR’s scope.
2. Lines 72–77 mention that prior datasets require hand-picking and are small. But I cannot understand why this new dataset and format can bridge the gap, why they do not need hand-picking, and how they ensure high-quality data.
3. Lines 62–68 show that STEP requires CAD kernels for processing, and version incompatibility is an issue. Why not just convert to the newest version so it is general and does not need the Better STEP format?
4. The paper claims the proposed format and dataset “bridge the gap,” enabling LLMs to directly generate B-rep models, but it provides no supporting experiments.

**Questions:**

See weakness

---

> ### Author Response · Authors · 2025-11-20
>
> We thank the reviewer for their thoughtful feedback, and we are happy to address the raised concerns in detail below.
> ### ICLR’s scope
> Our intention with this submission is precisely to contribute to the “datasets and benchmarks” and “infrastructure, software libraries” areas that are explicitly listed as relevant subject areas in the ICLR call for papers. More specifically, our paper is not only a raw data release:
> - It introduces a new representation (Better STEP) that makes B-reps directly consumable by standard ML pipelines, and
> - It provides a large-scale dataset and accompanying library that removes the current practical barriers to learning in the B-rep domain.
>
> ### Hand-picking
> Lines 72-77: SolidGen (Jayaraman et al., 2023) explicitly notes that “publicly available datasets for B-rep solid models are limited compared to other 3D representations” and evaluates on DeepCAD, a filtered subset of ABC. Feng et al. (2023) likewise remark that “there is no publicly available dataset for feature edge detection tasks on point clouds,” and therefore synthesizes a dataset from ABC meshes, noting that manually labeling edges on point clouds is tedious and does not scale. Fu et al. (2023) observes that the ABC dataset lacks the annotations needed to learn Bézier decomposition on point clouds, and thus introduce a non-trivial preprocessing pipeline with CGAL and OpenCascade to generate suitable labels. In contrast, our dataset is obtained by automatically converting all ABC and Fusion 360 STEP files, without any manual category or shape selection or task-specific annotation pipelines. This removes the need for hand-picking and one-off conversions and yields a much larger and more diverse B-rep corpus than the curated or heavily pre-processed subsets used in prior work. Based on our review of recent CAD learning literature, we identified a clear need for an ML-ready, large-scale, high-fidelity B-rep dataset and designed our contribution explicitly to address this gap. The aforementioned works, and similar future efforts, could directly leverage our format and accompanying library to extract as much task-specific data and supervision as needed, without bespoke preprocessing pipelines or manual curation.
>
> ### How we ensure high-quality data
> Our goal is not to create new CAD content from scratch, but to provide a faithful, ML-ready representation of existing, widely used CAD datasets. The original ABC and Fusion 360 Gallery repositories already enforce basic CAD validity and correspond to realistic engineering designs. Our conversion pipeline preserves the B-rep exactly: we store all parametric curves and surfaces, as well as the explicit topology, in a lossless HDF5 representation (Sec. 3).
>
> ### Why not use the latest version of CAD kernels
> Lines 62–68: Version incompatibility is only one of the reasons STEP is impractical as an ML data format. STEP is a low-level, highly verbose, and conceptually complex standard that was designed to be consumed by CAD kernels, not by researchers or typical ML tooling. In practice, a raw STEP file is almost unreadable without a kernel such as OpenCascade (or, more commonly, commercial software), and even worse, the seemingly simple tasks like, “iterate over all faces”, “get the NURBS surface for this face”, “find the boundary edges”, require navigating a dense standard and non-trivial C++/Python bindings. This is why, to the best of our knowledge, ML papers do not train directly on STEP: they always hide it behind a preprocessing stage that converts STEP into meshes, point clouds, or task-specific labels via a CAD kernel. In our work, we make this complexity explicit and handle it once: we use OpenCascade, a free and widely used kernel, to parse STEP and then export all curves, surfaces, and topology into a clean HDF5 schema (“Better STEP”). These HDF5 files can be inspected and loaded with standard tools on any platform, with no CAD software, no licenses, and no STEP expertise. Simply upgrading everything to the latest STEP version would not solve these issues; it would still leave users dependent on a CAD kernel and the full STEP stack at training time. Our format is specifically designed to remove that barrier and make B-reps genuinely ML-ready.
>
> ### LLMs and generative models
> The mention of LLMs in the conclusion was intended as a forward-looking motivation, not as a claim that we already implement such pipelines; our goal was to highlight a potential use case that our dataset and representation could enable. We are aware of existing approaches that use LLMs for CAD generation or modification, but they operate on sequences (e.g., command histories) or 2D sketches rather than directly on B-reps. If the reviewer is aware of such work, we would be very grateful for pointers and would be happy to showcase it using our interface and to include it in the paper.

---

### Official Review · Reviewer_3b7z · 2025-10-27

**Soundness:** 3
**Presentation:** 3
**Contribution:** 3
**Rating:** 6
**Confidence:** 4

**Summary:**

This paper proposes "Better STEP"—an open format equivalent to B-rep and its accompanying Python libraries. This aims to bypass the dependency of proprietary CAD kernels on STEP, preserving fully parameterized geometry and topology for direct access and sampling by ML. The authors convert large datasets such as Fusion 360 and ABC to this format and demonstrate data generation and evaluation pipelines for downstream tasks such as normal estimation, denoising, reconstruction, and segmentation. They reproduce close to published results without fine-tuning, demonstrating the versatility and practicality of the format and toolchain.

**Strengths:**

Bypassing proprietary kernels and version incompatibilities, it directly provides B-rep equivalent representations that can be consumed by ML frameworks;

Provides a clear hierarchical structure (geometry/topology/mesh), standardized APIs (sampling, normals, curvature, topology traversal, etc.), and reports statistics such as conversion and failure rates;

The same interface can generate data for multiple types of downstream tasks, with reasonable example coverage, showing plug-and-play support for existing methods.

**Weaknesses:**

The main problem is the insufficient demonstration of reproducibility and usability: there is currently no external demo, sample data or minimum runnable script, and reproduction must wait for formal acceptance and release, which has a high threshold; there is a lack of online browsing/interactive examples to demonstrate "ease of use" (such as visualization of typical B-rep, one-step sampling/export of point cloud); insufficient display of generated scenes - although the paper discusses the potential of LLM/CAD generation, it lacks specific case studies or quality inspection indicators from point cloud to B-rep, or from text/code to B-rep via GPT; the legal and licensing aspects are not detailed enough (the redistribution terms after Fusion 360/ABC conversion, the scope of subset disclosure, commercial use restrictions, etc. need to be clarified).

**Questions:**

Is better to provide a "minimal working subset" (e.g., HDF5 for 50–100 models, corresponding visualization and sampling scripts, and pre-generated downstream task examples) during the review period to verify usability? Or provide an online demo.

Regarding the promise of "generating B-reps from point clouds/from LLMs," can you provide at least a few end-to-end or overfitting examples to demonstrate how the data/interface actually facilitates the generation tasks?

---

> ### Author Response · Authors · 2025-11-20
>
> We thank the reviewer for their thoughtful feedback, and we are happy to address the raised concerns in detail below.
> ### Concern regarding reproducibility, usability, and licensing
> The concerns about reproducibility and usability are largely a consequence of the double-blind nature of the ICLR review process, rather than a lack of available resources. In practice, the converted dataset is already publicly available on persistent storage, consists of several terabytes of data, and is ready to download. The accompanying Python library is likewise already online and distributed via pip, so it can be installed and used with standard tooling. We have also meticulously designed a license to ensure compatibility with the original Fusion 360 and ABC licenses, and this license text is provided alongside the released data. The only reason the submission does not contain explicit links, package names, or other identifying information is strict adherence to ICLR’s double-blind policy. When anonymity is no longer required, we can simply restore these concrete references so that reviewers can directly verify the public availability and ease of use of the dataset, code, and license. If the conference policy allows it, we would be happy to provide reviewers with links to a small sample subset of the dataset and the corresponding library/demo scripts (or to add such links) so that usability can be verified directly.
> ### LLMs and generative applications
> The mention of LLMs in the conclusion was intended as a forward-looking motivation, not as a claim that we already implement such pipelines; our goal was to highlight a potential use case that our dataset and representation could enable. We agree with the reviewer that end-to-end, or even small illustrative, generation examples would be very valuable to showcase future directions. However, to the best of our knowledge, there are currently no existing methods that provide full point-cloud-to-B-rep or LLM-to-B-rep generation pipelines on an open B-rep dataset like ours, we are aware of existing approaches that use LLMs for CAD generation or modification, but, they operate on sequences (e.g., command histories) or 2D sketches rather than directly on B-reps. If the reviewer is aware of such work, we would be very grateful for pointers and would be happy to showcase it using our interface and to include it in the paper.

---

### Official Review · Reviewer_jF6q · 2025-10-31

**Soundness:** 3
**Presentation:** 2
**Contribution:** 2
**Rating:** 2
**Confidence:** 4

**Summary:**

The authors propose BetterSTEP, a dictionary-like, half-edge-based, open-source format for representing B-Reps (Boundary Representation) of 3D CAD models, which enables easier integration into standard machine learning pipelines. They provide an OpenCascade-based conversion library that can convert STEP files into the BetterSTEP format, as well as another library called 'abs' for processing and querying B-rep data in the BetterSTEP format. The 'abs' library provides utilities for querying and processing both the geometry and the topological structure.  They provide a dataset formed by combining the ABC dataset and the Fusion 360 dataset and converting to the BetterSTEP format using the proposed conversion library. The authors sample point clouds from the dataset and utilize these point clouds as inputs to existing deep learning models for four different tasks: normal estimation, denoising, reconstruction, and segmentation. On these tasks, they provide quantitative and qualitative results, finding equivalent or slightly reduced performance compared to the original reported results (without fine-tuning or training the models). Overall, the paper proposes a new representation for B-Reps and CAD models that is open-source and equivalent to the original B-Reps while allowing easier integration with SoTA deep learning pipelines.

**Strengths:**

- A standard format for representing B-Reps from different sources in a consistent manner that can be easily utilized in Python, and a dataset in this format, is beneficial for the research and development of new approaches, simultaneously allowing for better and more consistent benchmarks and evaluation of these approaches.
- The provided format is independent of the original (commonly proprietary) CAD file format, which allows combining different datasets. In addition, the proposed 'abs' library allows querying and processing the inputs using the proposed format, reducing the requirement for different data processing pipelines.

**Weaknesses:**

- Code listings are not clear. Listings 4 and 5 are used to replace the compute_labels function from Listing 2; however, inconsistent return formats (1/0 vs 1/None) are used between different examples. Additionally, Listing 3 does not provide any meaningful/helpful information. Pseudocode detailing how read_meshes/get_mesh worked would be more useful than the current provided code. In general, I think the provided code could be clearer, and more details could be provided in addition to the very high-level usage examples. Moreover, the examples could be made more compact; for instance, by removing the multiple empty lines from Listing 2, and utilizing abs.function_name instead of having similar imports in different Listings.
- The contributions are not clear. The main contribution regarding the dataset is conversion, which is also repeated as part of the libraries as a conversion library. The provided dataset is a combined version of the ABC and Fusion 360 datasets, converted to the proposed format by parsing the STEP files with OpenCascade, extracting the geometric and topological information, and saving them into the proposed dictionary-like format. Based on this, it is unclear what additional improvements are provided beyond the geometric and topological information provided by OpenCascade. Moreover, failure percentages are reported for the meshing algorithm of OpenCascade, along with some failure case examples; however, the reasons for these failures are not discussed beyond the reported percentage failure rates. In addition, it is not clear whether StepToHDF5 should be considered a library here instead of a function handling inputs as part of the provided 'abs' library, which handles reading and navigating the HDF5 output files. Overall, the paper would benefit from improved structuring and clearer explanations regarding its contributions.
- Even though the format is independent of the conversion process, only OpenCascade is utilized in the paper to convert STEP files to the proposed format. However, OpenCascade also has its own BREP format. Currently, the paper does not discuss the advantages/strengths of the proposed format over this format, which would strengthen the claims.
- Limited comparisons and modality in experiments. The paper claims that the proposed format for B-reps would make them easier to use in standard machine learning frameworks. However, only point-cloud input models are considered. Moreover, only performances on existing models are reported, using point cloud inputs sampled from the proposed format and library, with no comparison to inputs sampled from B-rep or meshes directly. In these cases, the original performances are not reported, making it hard to evaluate the provided experiments. Additionally, there are no examples of training or fine-tuning. In my opinion, these provide very limited support to the claims. The paper would be significantly improved with the incorporation of geometric/topological deep learning models and/or mesh-based models.
- The paper claims that the proposed dataset and format will "bridge the gap, enabling LLM to generate complex B-rep models fully"; however, no LLM-based experiments are provided to support this claim.

**Questions:**

My primary concerns are the unclear claims and contributions, as well as the limited experimental results. Could the paper clarify the contributions regarding the dataset, libraries, and the format? Additionally, I am not certain how the numerical results in the provided experiments were evaluated or what the expected numbers were, as there are no proper comparisons to the base work/other works; could you provide a detailed explanation regarding these?

---

> ### Author Response · Authors · 2025-11-20
>
> We thank the reviewer for their thoughtful feedback, and we are happy to address the raised concerns in detail below.
> ### Code listings
> The `compute_labels` function in Listings 4 and 5 are user-defined functions. The library does not fix their return type or values: the callback returns some data for the sampled points, or `None` to skip the current entity. We intentionally used different outputs to illustrate this flexibility of the library.
> The code snippets in the paper are minimal usage sketches, not a full API specification. The library exposes the full B-rep geometry and topology, including all parametric surfaces and curves with their derivatives; implements interior sampling; supports Poisson downsampling; and provides a Python API for high-level operations. Due to space limits, we show only the core usage pattern rather than the full API details.
> ### Contribution Clarification
> Our goal is not to re-implement a CAD kernel, but to expose the geometry and topology that currently lives only as opaque, in-memory OpenCascade objects into a documented, stable, ML-oriented format that can be used without a CAD kernel at training time. OpenCascade is only the parser; Better STEP is the standardized, reusable representation that the ML community can store, share, and build upon.
> Our contribution goes beyond “just conversion”:
> - We define a fully specified, cross-platform HDF5 schema with half-edge topology and explicit links between geometry, topology, and per-face meshes, independent of any kernel.
> - We provide two libraries: STEPTOHDF5, which converts STEP files into one consistent format, and ABS, which offers high-level ML-facing operations (topology navigation, sampling, normals, curvature, labels, mesh extraction, etc.) not available in OpenCascade.
> - We apply this pipeline to convert >1M models and show that off-the-shelf models for normals, denoising, reconstruction, and primitive segmentation can operate directly on our format.
>
> ### Comparisons
> For all use cases, we do not train or fine-tune on our dataset. Even in this “plug-and-play” setting, we obtain accuracies close to original papers, which indicates that our data faithfully preserve the underlying geometry. Our main claim is that existing models can use our representation without any modification, and these experiments directly support that.
>
> |Method |Setting|Original| Ours|
> |-|-|-|-|
> | BPNET| Accuracy | 96.83%| 85.78%|
> | | Avg. primitives per shape| 19.17| 23|
> | NKSR| σ = 0, F-score| 93.2| 84.62|
> | | σ = 0, Chamfer dist| 3.68| 3.39                              |
> | DeepFit | PGP @ 5° / 10° / 30°                    | 70 / 80 / 90 %                    | 65.73 / 74.95 / 90.28 %           |
> | PathNet | MSE @ 0.5 / 1 / 1.5% noise | 29.03 / 29.44 / 30.03              | 33.90 / 34.07 / 35.79             |
>
>
> ### Point clouds and missing LLM
> Point clouds are currently the most common input. Since this is a dataset and representation paper, proposing new topological architectures or LLM models that use this additional information is outside the scope of the current work, and the mention of LLM in Conclusion is meant as a forward-looking motivation rather than a claim that we already have such an LLM.
> We are aware of existing approaches that use LLMs for CAD generation or modification, but, to the best of our knowledge, they operate on sequences or 2D sketches; this is different from what our dataset provides, as we expose explicit B-rep geometry and topology in a standard ML format so that future models can work directly at the B-rep level. If the reviewers are aware of prior work that we would greatly appreciate pointers and would be happy to include and discuss them in the paper, and to add comparisons or additional experiments where feasible.
> ### Meshes
> We report OpenCascade meshing failures for two reasons. (1) Our dataset also includes the meshes, but not every STEP file yields a mesh (e.g., Fig. 2). (2) Many existing works using the ABC dataset rely exclusively on meshes, and therefore implicitly skip all models for which meshing fails. By reporting these failures, we highlight that relying only on meshes introduces a dataset bias and systematically excludes complex shapes.
> Regarding why OpenCascade fails, meshing STEP files is still an open problem; failures typically arise from the STEP geometry itself or limitations of current meshing pipelines. This is precisely why we argue that using our representation is more robust for downstream learning tasks.
>
> ### OpenCascade
> We used OpenCascade solely as a conversion backend because it is the only open-source CAD kernel that can robustly parse large-scale STEP collections.
>
> ### StepToHDF5
> Our intention is for the community to build additional CAD datasets in the future. StepToHDF5 is meant as a utility that helps convert raw STEP files into our HDF5 format; we will clarify this to avoid confusion about whether it is a standalone library or a helper function.

---

### Official Review · Reviewer_oX1K · 2025-10-31

**Soundness:** 2
**Presentation:** 2
**Contribution:** 1
**Rating:** 2
**Confidence:** 5

**Summary:**

Paper prosed better step, an open source cad brep storage format to support further research in cad ML community. CAD models from abc and fusion360 are converted to hdf5 format. Authors provide python conversion script and other functions (e.g sampling normals).

**Strengths:**

HDF5 is a well supported and common format. Authors demonstrate some builtin functions using their python library. Converting different dataset into this version might be beneficial to open source? Although I have some difficulties understsanding what open source mean in this context. My understanding is that this python script is still built upon opencascade.

**Weaknesses:**

Paper lacks contribution in dataset and benchmark. No new dataset is introduced. Authors merely converted abc and fusion360 datasets into their "better step" format. Also there is no new data structure or more ML-friendly data representation for training. BRep is still represented by parametric faces, shells, edges, and vertices but with their parameters stored in hdf5 format. The topology is also still a linked list (top-down now). To me this doesn't really make the data any more "ML-friendly" than the standard STEP format.

**Questions:**

How is the format "better" besides stored as hdf5 and has some python library functions. The rebuild seems to be slightly better but not signifiantly different from opencasde.

STEP is very efficient and the most common data format for CAD model sharing. How does hdf5 compared to STEP in terms of storage size or reading / loading speed.

Would abc data converted to "better step" format help improve CAD generation?

---

> ### Author Response · Authors · 2025-11-20
>
> We strongly disagree with the claim that STEP is ML-friendly. If it were, existing ML papers on CAD would train directly on raw STEP files. Based on our literature review, they either hand-pick small subsets, use ad-hoc intermediate formats (e.g., meshes), or construct small datasets using different formats tailored to specific tasks. This repeated effort on its own is enough evidence that STEP, as it stands, is not practical.
>
> To further justify our claims, we refer to the following papers just to name a few:
>  >The primary limitation of these datasets is their lack of parametric and topological features of curves and surfaces commonly referred to as boundary representation (B-Rep), … Advancing across various tasks in GDL, and effective training of deep learning models, necessitate large parametric CAD data collections. The existing parametric CAD datasets with B-Rep data are limited in size and insufficient to meet these demands.
> [Heidari, Negar, and Alexandros Iosifidis. "Geometric deep learning for computer-aided design: A survey." IEEE Access (2025)]
>
> > Because there is no publicly available dataset for feature edge detection tasks on point clouds, we generate our dataset as synthesized from 3D models. We first collect 3D meshes from the ABC dataset, the largest publicly available dataset of CAD models, as the ground truth surface. The problem is that manually labelling edges on point clouds is tedious, and manual labelling makes it difficult to obtain large amounts of data.
> [Feng, Yi-Fei, et al. "Deep shape representation with sharp feature preservation." Computer-Aided Design 157 (2023): 103468]
>
> > We evaluate our approach on the ABC dataset. However, the ABC dataset does not have the annotations to learn Bézier decomposition on point clouds. Therefore, we do a pre-processing step. Specifically, we utilize the CGAL library and OpenCascade library to perform Bézier decomposition on STEP files directly and perform random sampling on the surface to obtain the following label
> [Fu, Rao, et al. "BPNet: Bézier Primitive Segmentation on 3D Point Clouds." arXiv preprint arXiv:2307.04013 (2023)]
>
> > Publicly available datasets for B-rep solid models are limited compared to other 3D representations
> [Jayaraman, Pradeep Kumar, et al. "Solidgen: An autoregressive model for direct b-rep synthesis." arXiv preprint arXiv:2203.13944 (2022)]
>
> The core of Better-Step is a B-rep representation explicitly tailored for ML. All geometric entities (faces, edges, vertices), their parametric surfaces/curves, parameter domains, and derivatives are stored as dense numeric arrays with stable integer indices. Topology (face–edge, edge–vertex, etc.) is encoded as index arrays that plug directly into tensor/graph libraries and support easy batching. Surface and curve types are given by compact enums plus parameter tensors, making them directly tokenizable/embeddable for LLMs and graph networks. Loading a model in Python is a single h5py call, and iterating over faces/edges is just array slicing, no STEP parsing, no kernel calls. By contrast, STEP is a verbose relational text standard that only becomes useful after interpretation by a CAD kernel, with no standard way to obtain parametric patches or batched adjacency tensors, and frequent kernel-specific incompatibilities. Better-Step is precisely this missing ML-native B-rep data structure.
>
> ### Dependence on OpenCascade
> By “open source” we mean that the format specification, the library, and the converted dataset are all publicly released under licenses compatible with ABC and Fusion 360. Our pipeline intentionally uses OpenCascade as a backend to interpret STEP as our goal is not to re-implement a CAD kernel, but to extract the geometry/topology that currently lives as opaque in-memory kernel objects and convert it into a documented, stable, ML-facing representation that requires no CAD kernel at training time. In short, OpenCascade is the parser; Better-Step is the standardized, reusable output that the ML community can share and build on.
>
> ### Storage and speed
> A direct STEP vs. HDF5 read speed comparison is not meaningful because the formats serve different roles. STEP must be parsed through a CAD kernel just to access geometry/topology, whereas our HDF5 is a post-conversion, ML-facing representation that can be read directly. The STEP to HDF5 conversion is a one-time cost (for which we already report throughput); once the dataset is distributed, users never rerun this step. During training/evaluation, models only read HDF5, with I/O patterns similar to other large 3D ML datasets, so comparing STEP via kernel to already converted HDF5 is not an apples-to-apples benchmark.
> However, to address the reviewer’s concern: on Fusion Joint, reading HDF5 takes a few milliseconds on average (0.38s for a batch of 1,000 samples); on Fusion Assembly, 0.75s; and on ABC, which has the largest models, 3.57s. Average storage per model is 3.05 MB for Fusion Assembly and 16.86 MB for ABC.

---

### Meta-Review · Area_Chair_LR4L · 2025-12-19

**Summary:**

The reviewers primarily raised the following concerns:

1. The potential applicability to machine learning, manifested mainly in the lack of theoretical justification and implementation evidence.

2. The specific advantages of the proposed BetterSTEP format over STEP format are not sufficiently demonstrated.

3. Other issues, such as clarity of presentation, reproducibility, and intellectual property licensing.

**Reviewer Concerns:**

1. Regarding the claimed suitability for machine learning: The authors' experiments only demonstrate that the BetterSTEP format faithfully preserves geometric structure, but they provide no evidence of pre-training or fine-tuning results based on this format that would indicate improved machine learning performance. Additionally, the assertion that the non-sampled original geometric structure is suitable for LLMs lacks experimental support. The authors note the absence of comparable prior work as a reason for not providing baselines. In both the sampled point-cloud and non-sampled original-structure settings, the current experiments do not substantiate the claim that the BetterSTEP format is "better suited for machine learning." Consequently, I believe the reviewers' concerns on this point remain unaddressed.

2. Regarding the advantages of the BetterSTEP format over STEP ones: The authors argue that the primary benefits lie in standardization, ease of use, and greater suitability for machine learning. However, since the latter point is not supported by the provided experimental evidence, the overall advantages relative to existing format become considerably less compelling. Thus, the reviewers' concerns here also persist.

3. Other issues, including clarity of writing, reproducibility (e.g., sample data, demo scripts), and IP licensing (e.g., redistribution terms, commercial use restrictions): The authors have provided detailed and comprehensive responses, and I consider these concerns resolved.

**Reviewer Scores:**

I do not expect any reviewer to change their score. The reviewers' core concerns, applicability to machine learning and clear advantages over STEP format, remain unresolved.

---

### Decision · Program_Chairs · 2026-01-26

Reject